# Effects of a Cloth Panel Containing a Specific Ore Powder on Patients with Japanese Cedar Pollen Allergy During the Pollen Dispersal Season

**DOI:** 10.3390/jcm8122164

**Published:** 2019-12-06

**Authors:** Suni Lee, Shoko Yamamoto, Maiko Hamana, Hiroshi Okamoto, Tamayo Hatayama, Fukusou Danbara, Yoshio Fujii, Youichi Murakami, Takemi Otsuki

**Affiliations:** 1Department of Hygiene, Kawasaki Medical School, 577 Matsushima, Kurashiki City, Okayama 701-0192, Japan; slee@med.kawasaki-m.ac.jp (S.L.); s.yamamoto@med.kawasaki-m.ac.jp (S.Y.); tama@med.kawasaki-m.ac.jp (T.H.); 2Wadakohsan Corporation, 4-2-13, Sakaemachidori, Chuo-ku, Kobe City, Hyogo 650-0023, Japan; maiko-h@wadakohsan.co.jp (M.H.); danbara@wadakohsan.co.jp (F.D.); murakami@wadakohsan.co.jp (Y.M.); 3Cosmic Garden Co., Ltd., 1-2-25 Ima, Kita-ku, Okayama City, Okayama 700-0975, Japan; okamoto@cosmic-g.jp (H.O.); info@cosmic-g.jp (Y.F.)

**Keywords:** pollen allergy, natural ore powder, cloth, interior material, symptom relief, immunological effect

## Abstract

Pollen allergy remains a big problem in contemporary societies. We have shown in previous studies that a cloth containing a special natural ore powder (CCSNOP) is effective in relieving symptoms in patients with pollen allergies. However, in that study, subjects were exposed to CCSNOP for only one hour. In the present study, CCSNOP or control (non-woven cloth; NWC) panels were placed in the bedrooms of pollen allergy patients for two weeks during the pollen dispersal season in 2018, and the effects were investigated. Twenty-one subjects were exposed to CCSNOP panels and 10 subjects were exposed to NWC panels. Our investigations showed that use of CCSNOP resulted in relief of symptoms and reduced use of therapeutics. Moreover, the ratio of eosinophil count decrease during exposure was higher in the CCSNOP group. Furthermore, a formula for measuring various cytokines and other parameters was established and clearly showed a distinction between the CCSNOP and NWC groups. In this formula, Granulocyte Macrophage colony-stimulating Factor (GM-SCF), Interleukin (IL)-12p40, Immunoglobulin (Ig) G4 and eosinophil count were extracted. These results indicated that CCNSNOP has a beneficial effect on pollen allergy patients. Future studies shall engage in long-term monitoring of pollen allergy patients who will utilize this mineral powder for at least one year.

## 1. Introduction

Pollen allergies and hay fever, typically manifesting as allergic rhinitis and allergic conjunctivitis, represent seasonal allergic reactions that can limit the ordinary lives of individuals [1,2,3]. Patients with pollen allergies experience various nasal symptoms such as sneezing, runny nose, and stuffy nose, in addition to eye-related conditions such as redness, watery eyes, and itching [1,2,3]. In Japan, the number of people with allergic responses to Japanese cedar pollen has been increasing over the last two decades [4,5,6]. Additionally, some patients with pollen allergies can have adverse reactions to as many as 50 types of plants including cypress, rice, ragweed, and mugwort. The treatment of pollen allergy is generally limited to suppressing symptoms, with oral antihistamines, nasal drops, or eye drops typically being used as therapeutic strategies [7,8,9]. Although desensitization therapy has been tried [9,10,11], it has not been fully effective, and patients just wait for the pollen season to pass. Precautions can be taken to avoid or minimize exposure to pollen through decreasing outdoor activities, closing windows, operating central air conditioning systems with filter attachments, bathing and shampooing the body and hair well to remove attached pollens, and drying clothes well. Notwithstanding these precautions, patients with pollen allergies continue to suffer with symptoms, even in their own homes, during the pollen season [4,5,6].

The Cosmic Garden Co. Ltd. is a custom-home building company located in Okayama City, Japan. Some of the features of their homes include a modified “2 × 4” construction method for aseismic ability and the avoidance of chemical substances and smells when possible, as well as the construction of well-sealed, super-insulated, durable homes. As we previously reported, this company uses specific natural ore powder within wall materials when constructing homes. This natural ore is obtained near Aso Mountain, Kumamoto Prefecture, Kyushu Island, Japan, and is known to release far-infrared rays. The appearance of this natural ore is shown in Figure 1A and the chemical analyses of four main parts (shown in Figure 1A) of this natural ore examined by an X-ray fluorescent (KrF) method are shown in Figure 1B, as previously reported. The analyses revealed no remarkable characteristics. Additionally, the Cosmic Garden Co. Ltd. has incorporated this natural ore into the wall material of more than 200 homes. Home occupants have expressed improvements in allergic symptoms such as pollen allergy, bronchial asthma, and atopic dermatitis. Additionally, many occupants reported improved sleep quality. Moreover, the office space of this company includes this specific ore powder in the walls, and many customers have claimed immediate improvement of symptoms related to pollen allergy after entering this office space. 

In our previous study [12], we investigated the biological effects of cloth containing a specific natural ore powder (CCSNOP) on patients with pollen allergy in terms of symptoms, biological markers, mood state via a questionnaire referred to as Profile of Mood Status 2 (POMS2), stress marker using salivary amylase (sAmy), serum immunoglobulin (Ig) E specific to 33 antigens, peripheral blood counts including percentage of white blood cells, and 29 serum cytokines under conditions when subjects were exposed to CCSNOP for 60 min. For the control, subjects were exposed to non-woven cloth (NWC). From this previous study, some symptoms such as nasal obstruction and lacrimation improved, and POMS2 evaluations showed that patients were calmer following their stay with CCSNOP. However, relative eosinophil percentage, non-specific Ig E, epidermal growth factor (EGF), monocyte chemotactic protein-1 (MCP-1) and tumor necrosis factor-α levels were higher in subjects exposed to CCSNOP compared with NWC. However, in that previous study, it was not confirmed whether exposure to CCSNOP did indeed improve symptoms of pollen allergy, or whether the indoor air circumstances using CCSNOP affected the human immune system [12].

The present study was executed after Wadakohsan Corp. (a real estate business) began selling detached homes containing CCSNOP provided by Cosmic Garden Co. Ltd. In this study, CCSNOP or NWC (Figure 1C) panels (of indistinguishable appearance) were placed in the sleeping rooms of subjects (Figure 1D) with pollen allergy (self-reported) for two weeks at least one or more weeks after the beginning of the Japanese cedar pollen dispersal season. Subjects were required to keep a “pollen allergy diary” that included details of symptoms, medication employed, and other comments (Figure 2). Additionally, biological monitoring was performed one month prior to the anticipated pollen dispersal, immediately prior to and following panel placement, and two months after Japanese cedar pollen dispersal had ceased.

## 2. Experimental Section

### 2.1. Subjects

All 31 subjects were Japanese. Subjects were recruited following public announcement of the intended study. The subjects comprised 14 males (average age (AA) ± standard deviations (SD): 45.1 ± 14.1 years old) and 17 females (37.1 ± 11.0 years old), with the AA of the 31 subjects being 41.2 ± 12.8 years old. Eight subjects were living near Kobe City, Hyogo Prefecture, Japan. For them, the POMS2 questionnaire, sAmy sampling, and venous blood collection were done within rooms of Wadakohsan Corp. in Kobe City (Kobe venue). The other 23 subjects were living in Okayama Prefecture. Thus, sampling was performed in rooms at Okayama University in Okayama City, and rooms at Kawasaki Medical School in Kurashiki City, with both cities being in Okayama Prefecture (ca. 10 km apart; Okayama venues). The cities of Kobe, Okayama, and Kurashiki are in the western region of Japan and are ca. 120 km apart. However, dispersal of Japanese cedar pollens does not differ markedly between these regions according to weather forecast data and the Japanese Ministry of the Environment. The Kobe venue included eight subjects (Male:Female = 4:4), while the Okayama venues included 23 subjects (M:F = 10:13). All subjects self-reported pollen allergy to Japanese cedar with various typical symptoms. Subjects with well-controlled lifestyles possessing diseases such as hypertension, diabetes mellitus, respiratory diseases, and liver and kidney diseases were included in the study; however, patients with cancers, collagen diseases, and other severe diseases were omitted from the study. All subjects continued with their regular daily lives during the study without modification. 

The study was designed as a double-blind study. Thus, all researchers measuring POMS2, sAmy, and blood sampling were unaware of which panel (CCSNOP or NWC) each subject was being exposed to. Following the final sampling in June, 2018, the distribution of subjects within the CCSNOP and NWC groups was revealed. The Kobe venue had 2 males and 2 females with CCSNOP and 2 males and 1 female with NWC. The Okayama venues included 6 males and 10 females with CCSNOP and 4 males and 3 females with NWC. In total, 8 males and 13 females (21 subjects) comprised the CCSONOP group and 6 males and 4 females (10 subjects) comprised the NWC group. CCSNOP and NWC groups differed since we designed this study not only to compare the two groups but also to analyze changes in biological markers among subjects of the CCSNOP group. 

### 2.2. Ethical Matters

This study was approved by the Kawasaki Medical School Ethics Committee (Issue No. 2576, date of approval December, 12th, 2016). All subjects of this study were approached verbally and in writing, and those from whom written consent was obtained were recruited as subjects in this study. All methods used in this study were performed in accordance with the relevant guidelines and regulations shown by Kawasaki Medical School Ethics Committee as well as the Declaration of Helsinki. Additionally, the monitoring and audit for this study were performed by the Academic Research Organization (ARO), Center for Innovative Clinical Medicine, Okayama University, since favorable results may be used in the advertising of Cosmic Garden Co. Ltd. and Wadakohsan Corp. There were no problems with the monitoring and auditing of this study.

### 2.3. Study Design

As shown in Figure 3, the study was performed from January to June, 2018. The samplings (POMS2, aAmy, and blood collection) were performed four times. The first was early January, one month before pollen dispersal. The second was at the end of February, one or more weeks after the commencement of dispersal. The third was two weeks after the second time. During these two weeks each subject was exposed to CCSNOP or NWC panels that had been placed in their bedroom (Figure 1D). The last sampling was performed in early June, almost two months following the cessation of pollen dispersal. All dates are listed in Figure 3. One week prior to panel placement and during the 2-week period of panel exposure, all subjects kept a “pollen allergy diary” as shown in Figure 3. 

The degree of Japanese cedar pollen dispersal was similar in the Kobe and Okayama areas according to weather forecast data and the Japanese Ministry of the Environment. As shown in Figure 4, the amount of pollen during the period of CCSNOP or control panel installation did not differ significantly in either Okayama City (Okayama Prefecture) or Kobe City (Hyogo Prefecture) where the subjects lived. However, it greatly exceeded the 50/cm^2^ that the Ministry of the Environment of Japan stated as representing a “very high” pollen level.

### 2.4. Pollen Allergy Diary

As shown in Figure 3, all subjects kept a “pollen allergy diary” for 4 weeks. An ID number was assigned to subjects by the personal information manager, and was unknown to researchers. Diary entries contained the date, day of the week, and an assessment of the weather. Subjects also recorded any symptoms related to the nose or eye on a scale of 1 (none) to 5 (most severe). Furthermore, problems of daily life caused by pollen allergy was also ranked as 1 (none) to 5 (incomplete). Moreover, the use of medication (oral drugs, nasal drops and eye drops) was described. Other matters related to pollen allergy such as physical and mental condition were recorded in the “Others” category. To analyze symptoms “Before”, “During”, and “After” periods of panel exposure, as well as to compare CCSNOP and NWC groups, recorded scores of 1 to 5 were squared to clarify the differences (symptom-weighted score).

### 2.5. Survey of Mood (POMS2)

Investigation of mood was examined using POMS2 [14,15]. Although POMS2 is a descriptive questionnaire that conventionally determines the state of mood for the previous two weeks or so, in our study, subjects were asked to assess their mood state for that present moment.

The questionnaire comprised 30 questions, with answers that ranged in five stages from “not at all” to “very much”. The “feeling condition” was measured according to the following six scales:T-A:Tension-Anxiety (Tension-Anxiety), “Feeling tight/under tension” comprised five stages. The higher the score, the more nervous a subject feels.D:Depression (Depression), comprised five stages such as “feeling down or dark.” Higher scores indicate increased loss of confidence. A-H:Five stages such as Anger-Hostility and “Bad Mood”. A higher score indicates increased anger. F:Comprised five stages such as “Fatigue” and “I get tired”. A higher score indicates an increased feeling of being tired. C:Confusion, five items such as “Confusion”. A higher score indicates increased confusion and decreased lucidity. V:Vigor (Vigor), five items such as “Vivid”. As this item is a positive item, unlike the other five scales, a lower score indicates that the activity is lost.

Total Mood Disturbance (TMD) was also calculated. This is a comprehensive expression of negative mood state (formula for calculation shown below). The higher the main score, the more negative the mood; it was determined as follows:TMD = {(anger-hostility) + (confusion-embarrassment) + (depression-depression) + (fatigue-lethargy) + (tension-anxiety)} − (vigor-vitality)

### 2.6. Saliva Amylase

Subjects first rinsed their mouths before sampling, completed the POMS2 questionnaire, and then sAmy measurements were undertaken [16,17]. After placement of the dedicated stick (Nipro saliva amylase monitor chip Product Code 59-010) under the tongue for 30 seconds, the concentration of amylase was measured using a Salivary Amylase Monitor® (Nipro Corporation, Osaka, Japan) according to the manufacturer’s instructions. The results were recorded on a special form.

### 2.7. Blood Sampling

Fifteen milliliters of peripheral venous blood was taken from subjects and the following parameters were measured:General condition: blood chemistry including liver (AST, ALT, and γGT) and kidney (BUN and creatinine) functions, blood sugar, HbA1c and lipids (including LDL and HDL cholesterols and triglyceride), and peripheral blood counts including white blood cell fraction (granulocytes, eosinophil, basophil, monocytes, lymphocytes, and others) were recorded.Immunoglobulins (Igs): Ig G, A, M, and non-specific Ig E, as well as Ig G4 since Ig G4 is related to food allergy. More recently, Ig G4 has been considered to be associated with IG G4-related diseases, which are systemic diseases characterized by elevated serum IgG4, IgG4-positive plasma cell infiltration into the affected tissue, and fibrosis [18,19,20].Multi-antigen specific Ig E (33 types): This measurement kit typically included antigens for cedar, cypress, house dust, dermatophagoides pteronyssinus, and others. Measurements 1 to 3 above were performed in a clinical laboratory testing company (BML Inc., Shibuya-ku, Tokyo, Japan).Cytokines: Twenty-nine types of cytokines were measured using a Luminex Cytokine 29-Plex Human Cytokine/Chemokine Panel (HCYTMAG-60K-PX29, Merck Millipore, Billerica, MA). The cytokines included in this kit were EGF, eotaxin, granulocyte-colony stimulating factor (G-CSF), monocyte/macrophage-CSF (M-CSF), interferon (INF)-α2, IFN-γ, interleukin (IL)-10, IL-12p40, IL-12p70, IL-13, IL-15, IL-17, IL-1 receptor antagonist (IL = 1ra), IL-1α, IL-1β, IL-2, IL-3, IL-4, IL-5, IL-6, IL-7, IL-8, interferon γ-induced protein 10 (IP-10, also known as C-X-C motif chemokine 10 (CXCL10)), MCP-1, macrophage inflammatory protein ((MIP)-1α, also known as chemokine (C-C motif) ligand 3 (CCL3)), MIP-1β (also known as CCL4), tumor necrosis factor (TNF)-α, TNF-β, and vascular endothelial growth factor (VEGF). Some cytokines showed less than the measurement limitation. In these cases, for convenience, a value of one tenth of the lower limit of the measurement was substituted and utilized for statistical analyses [12,21,22].

### 2.8. Statistical Analyses

Statistical analyses were performed using SPSS version 22 (IBM, Chicago, IL, USA) or Microsoft Office Excel 2013 (Microsoft Japan, Tokyo, Japan). A significant difference was shown if the degree of risk was less than 5% (*p* < 0.05). To compare the symptom-weighted scores of CCSNOP (ore) and NWC (control; ctrl), POMS2 score analyses, sAmy concentrations, and absolute eosinophil counts, and for a comparison of values of the panel-prediction formula in subjects with CCSNOP and NWC, two-way analysis of variance (two-way ANOVA) and Student’s *t*-test (bilateral) were applied. Additionally, the use of allergy-related medicines by subjects and changes in absolute eosinophil counts were assessed by a χ-squared test. Regarding generation of the formula used to predict subjects who were exposed to CCSNOP or NWC, multiple regression analysis was performed. After obtaining the formula, a receiver operating characteristic (ROC) curve was also generated.

## 3. Results

### 3.1. Symptoms Related to Pollen Allergy

Changes in symptom severity are shown in Figure 5. Regarding stuffy nose, subjects with NWC (ctrl) showed increased severity in the “During” and “After” periods. However, subjects with CCSNOP (ore) revealed an increase in the severity of this symptom only in the “After” period. Although cedar pollen had been dispersed in the “Before” period, the increase in severity of this symptom in the control group seemed to suggest a natural course of pollen allergy. However, an increase in the severity of this symptom was not observed for the CCSNOP group in the “During” period (Figure 5A).

In terms of eye redness, both groups showed increased severity in the “After” period. Thus, there were no differences between the CCSNOP and NWC panel groups (Figure 5B).

As shown in Figure 5C, an assessment of the difficulty of everyday life showed similar results. Subjects with NCW (ctrl) showed increased difficulties in the “During” and “After” periods, while subjects with CCSNOP revealed increased difficulties only in the “After” period.

Figure 5D shows the results of a χ-squared test regarding days of use of medicines (general medicine, nasal or eye drops). Figure 5E shows the results of a χ-squared test regarding days of use of medicines only taken internally. Both results showed that subjects with CCSNOP (ore panel) had lower usage compared with NCW subjects (control panel).

All of these findings indicated that CCSNOP reduced the severity of symptoms associated with cedar pollen allergy. In particular, stuffy nose and difficulty in daily life improved during the installation period. In addition, the CCSNOP environment has led to reductions in medication. These points should be emphasized.

### 3.2. Changes in Mood and Stress

Figure 6A shows the changes in TMD as measured by POMS2 in individual subjects. There was no regular pattern observed of changes in TMD. A comparison of CCSNOP (ore) and NWC (ctrl) subjects was made and a time-course of the sampling time was assessed. However, no significant changes were observed (Figure 6B). Additionally, changes in sAmy, employed as a stress marker, are shown in Figure 6C. Again, no significance changes were observed between CCSNOP (ore) and NWC (ctrl) subjects. Although an assessment of the difficulties of everyday life indicated an improvement in CCSNOP subjects, mood and stress markers remained unchanged.

### 3.3. Specific Ig E and Eosinophils

Figure 7A,B show a stacked chart distribution of Japanese cedar specific Ig E classes in CCSNOP (ore panel) and NWC (control panel) subjects. Class 0 to 6 designations were as given by the BML Ltd. laboratory test. The classes and Ig E titers were as follows: class 0 (< 0.27 IU/mL), 1 (> 0.27 but ≤ 0.5), 2 (> 0.5 but ≤ 1.80), 3 (> 1.80 but ≤ 7.05), 4 (> 7.0 but ≤ 517.35), 5 (> 17.35 but ≤ 29.31) and 6 (> 29.31). Samples 1 to 4 represent sampling time 1 (January), 2 (before panel placement), 3 (just before panel removal), and 4 (almost two months after pollen dispersal). There were some differences regarding class distribution between CCSNOP (Figure 7A) and NWC (Figure 7B) subjects. The CCSNOP group included a higher class such as class 3 during the entire sampling period. There is a slight increase in class 3 in the control group, whereas no changes were observed in the CCSNOP group at sampling time point 3 (just after panel placement). Additionally, both groups showed an increase in classes at sampling time point 4. The reason why these classes increased when pollen dispersal had ceased remains unclear. However, since both groups showed similar patterns, this pattern was not dependent on the absence or presence of panels.

Figure 7C shows the changes in absolute eosinophil counts in peripheral blood among CCSNOP (ore) and NWC (ctrl) groups. There were no significant differences observed in all sampling times within each group as well as between the two groups at individual sampling times. Moreover, the changes in peripheral eosinophil count from sampling time 2 (just before panel placement) to sampling time 3 (two weeks after panel placement) revealed no difference between the two groups (Figure 7D). However, a χ-squared test to evaluate the number of subjects that showed an increase or decrease from sampling times 2 to 3 indicated that CCSNOP (ore) subjects showed significantly higher rates of decreasing eosinophil count compared with NWC (ctrl) subjects (Figure 7E).

### 3.4. Cytokine Analyses and Generation of Prediction Formula

Twenty-nine types of cytokines were measured using a Luminex 29 Cytokine Plex Kit (Merck KGaA, Darmstadt, Germany). Although our previous study examined changes in individual cytokines during stays (with sleeping) under CCSNOP or NWC conditions, in the present study, our aim was to measure these cytokines in an effort to determine whether CCSNOP affected the immune system.

In Figure 8, 28 types of cytokines were measured in four blood collections, and the average values and standard deviations in the CCSNOP and control groups are shown. IL-3 was excluded because it could only be measured with one sample. Moreover, a comparison between each group by Student’s *t*-test revealed that the CCSNOP group showed a higher value than the control group in the first and second blood collection of EGF. However, these differences disappeared after the third and fourth blood collections, and we do not consider them to be significant differences in this study.

As shown in Figure 9, with the seven cytokines, changes over time were observed as the results of the first to fourth blood collection in the CCSNOP group or the control group. However, as shown in Figure 8, these cytokines are also actually measured, and no difference is observed between the two groups. Also, the significant difference over time may or may not be significant depending on the standard deviation. However, as a whole, both groups showed similar trends.

From the results of Figure 8 and Figure 9, it was not considered that the improvement in symptoms and medications observed in CCSNOP was the effect of CCSNOP on any single cytokine.

As mentioned above, stays (with sleeping) under CCSNOP conditions for two weeks resulted in a reduction in the severity of symptoms and a decrease in the use of allergy-related medicines. A formula was generated to predict subjects who stayed under CCSNOP conditions by employing various parameters shown in Figure 10A that included 28 types of cytokines (IL-3 was omitted since all subjects revealed IL-3 values that were lower than the lower limit of measurement) using multiple regression analysis. All data at sampling time 3 (just after placement of the panels for the 2-week period) were used.

As shown in Figure 10B, four parameters with constant terms were extracted. Thereafter, the formula generated predicting subjects who stayed under CCSNOP conditions was as follows (also shown in Figure 10C):Panel-prediction formula = 0.801 − 0.061 × GM-CSF (pg/mi) − 0.050 × IL-12p40 (pf/mL) − 0.004 × Ig G4 (mg/dl) + 0.001 Eosinophil count (/μL)

After generating this formula, the data associated with these parameters for all subjects were substituted into this formula and plotted for CCSNOP (ore panel) and NWC (ctrl panel) groups as shown in Figure 10D. The formula detected with significant difference which panel type each subject was subjected to during the 2-week period. Additionally, as shown in Figure 10E, the ROC curve highlighted the successful predictive capacity of this formula, with (sensitivity) and (1-specificity) values of 0.905 and 0.100, respectively.

## 4. Discussion

Pollen allergies continue to be an issue for those affected in contemporary societies [1,2,3]. Although various strategies have been employed to deal with this medical issue, such as desensitization therapy [9,10,11], many patients simply endure by taking medicines that help to relieve the symptoms until the end of the pollen dispersal season. Most of the drugs only provide symptomatic relief, and in the end, the only recourse available to many patients is to try and avoid exposure [4,5,6].

We investigated whether the residential environment could induce some kind of pollen allergy relief. Cosmic Garden Co., Ltd. has been selling detached houses in which powder derived from ore collected near Aso Mountain, Kumamoto Prefecture, Kyushu Island, Japan, is mixed with interior wall materials. Anecdotal evidence suggested that home occupants experienced relief of symptoms related to pollen allergy. Therefore, in our previous study, we investigated the effects of mineral-containing (CCSNOP) or control (NWC) panels on pollen allergy patients every two weeks for one hour. The severity of symptoms, changes in eosinophils, cytokines, mood (measured by POMS2), stress markers measured by sAmy, and blood samples were investigated [12]. It was found that the severity of symptoms improved with CCSNOP. Furthermore, eosinophils increased slightly but significantly in the CCSNOP group [12]. Although changes in certain cytokine levels differed between the CSNOP and NWC groups, the biological significance of this finding remains to be determined [12]. Since it seemed that CCSNOP had a demonstrably positive effect on pollen allergy patients, it was thought that an investigation comprising longer-term exposure to CCSNOP, including during the pollen dispersal season, would be instructive.

Therefore, in the present study, we decided to examine various biological indicators along with symptoms in subjects with Japanese cedar pollen allergy who were exposed to CCSNOP or NWC panels in their bedroom.

The results of this study showed that CCSNOP alleviated the symptoms of pollen allergy and that the use of medicines decreased in the CCSNOP group. Inspection of the “pollen allergy diaries” revealed that even difficulties of daily life were reduced in the CCSNOP group, although the TMD (measured by POMS2) and the degree of stress (measured by sAmy) showed no differences between the CCSNOP and NWC groups.

However, blood sample analyses indicated that the absolute number of eosinophils tended to be lower in the CCSNOP group compared with the NWC group during the pollen dispersal season. 

Cytokines, especially MCP1, IP10, CXCL10/IP-10, CCL4, and CCL3, which are related to the onset of hay fever and the mechanism of symptom appearance, are shown in Figure 8. CCSNOP and control groups did not show significant changes. As discussed below, a formula was established to extract and detect subjects belonging to the CCSNOP group on the basis of these cytokines. These results suggest that this mineral powder panel may not have a direct effect on the pathogenesis of hay fever allergies, but may have an effect on secondary parts such as the appearance of symptoms. 

Furthermore, when many cytokines were measured and a formula for detecting CCSNOP subjects was generated along with other indicators, GM-SCF and IL-12p40 were extracted along with eosinophil count and Ig G4 values. These two cytokines are of greater importance. Additionally, when using this formula, it was revealed that blood sampling detected a significant difference in subjects exposed to CCSNOP panels for two weeks.

GM-CSF is known as a cytokine that induces differentiation of pluripotent hematopoietic stem cells into granulocyte-monocyte cells [23,24]. In relation to pollen allergy and GM-SCF, this cytokine has been reported to increase with IL-33 and IL-25 in animal models, and following the activation of neutrophils and antigen-specific T cells [25]. Additionally, one report has shown that IL-33 is activated by GM-CSF and IL-8 in the nasal mucosal epithelium of allergic rhinitis [26]. Moreover, some reports have indicated that production of GM-CSF was activated in patients with seasonal allergic rhinitis (possibly caused by pollen) [27].

Considering these reports and the prediction formula generated in this study, GM-CSF was extracted with a negative coefficient in the prediction formula, and could be utilized to detect a person staying under CCSNOP conditions. Therefore, CCSNOP may act on subjects by improving the pathophysiology of pollen allergy even at the cytokine level.

IL-12p40, along with IL-12p70, is a component of IL-12. IL-12 is well-known as a cytokine responsible for induction into Th1 cells together with IFN-γ [28,29]. Therefore, since a state with high IL-12 induces differentiation to Th1 rather than Th2, it might be that patients with pollen allergy possess reduced IL-12 production. However, IL-12p40 was also extracted with a negative coefficient in our formula. Although interpretation of this finding seems difficult, it may be that subjects exposed to CCSNOP experience decreasing levels of IL-12 over the 2-week period of stay, since they showed a reduction in the severity of symptoms during that period.

The formula generated in this study also indicated that IgG4 was extracted with a negative coefficient, even though the importance was less than that of the cytokines. Recently, IgG4 has been implicated to play a role in various diseases that have shown elevated serum IgG4, IgG4-positive plasma cell infiltration into affected tissue, and fibrosis [18,19,20]. However, IgG4 was initially considered in relation to the allergic condition. Production of IgG4 is induced by IL4 and IL-13, which are Th2-type cytokines mainly involved in allergic reactions under antigen stimulation. Thus, the negative coefficient of IgG4 in the generated formula may indicate a reduction in allergic reactions in subjects exposed to CCSNOP conditions.

In our previous observational study, we were able to confirm that the symptoms of hay fever improved in the 1-hour CCSNOP environment, but that was a unique environment of only 1 hour. While the subjects were living in a normal period, the effect was observed by setting up a CCSNOP environment in the bedroom at home, and as a result, symptoms were alleviated and medication was avoided. In addition, although there was no direct effect on individual cytokines allegedly associated with allergies, it is almost certain that it has an effect and that it alleviates allergic symptoms, and the mechanism is almost unknown at present, probably due to the decrease in pollen itself in the air. Whether the changes in the immune system that are shown in the current study are direct effects of the of CCSNOP or whether relaxation resulted in changes in such symptoms is, at present, unknown. 

There were no adverse effects in any of the subjects exposed to CCSNOP conditions in this study. Additionally, Cosmic Garden Co. Ltd. has sold more than 200 homes with powder or with this specific ore and no adverse effects have been reported.

Originally, it would be best if the effects of CCSNOP could be verified using animal models. However, it is very difficult to create a situation where the animal model is exposed to the environment by CCSNOP or NWC while building and using an animal model of hay fever in our laboratory. In future, it will be necessary to observe such a situation through some joint research. However, occupants of CCSNOP-containing detached homes sold by Cosmic Garden Co., Ltd. have already provided anecdotal reports of hay fever symptom relief. Consequently, an impact following the use of this specific ore could be expected.

The limitation of this study was that it comprised a small number of subjects and all results were extracted by comparing the CCSNOP group with the NWC group. The changes over time in this group did not yield a very definitive finding. However, we believe that exposure to CCSNOP reduced allergy-related symptoms in addition to affecting certain biological reactions in pollen allergy patients. Long-term monitoring of individuals living in homes containing this ore powder as part of the inner wall material may yield more information regarding the effects of CCSNOP on patients with pollen allergies.

## 5. Conclusions

Our investigations showed that use of CCSNOP resulted in relief of symptoms and reduced use of therapeutics. Moreover, the ratio of eosinophil count decrease during exposure was higher in the CCSNOP group. Furthermore, a formula for measuring various cytokines and other parameters was established and clearly showed a distinction between the CCSNOP and NWC groups. In this formula, GM-SCF, IL-12p40, IgG4, and eosinophil count were extracted. These results indicated that CCNSNOP has a beneficial effect on pollen allergy patients. Future studies shall engage in long-term monitoring of pollen allergy patients who will utilize this mineral powder for at least one year.

## Figures and Tables

**Figure 1 jcm-08-02164-f001:**
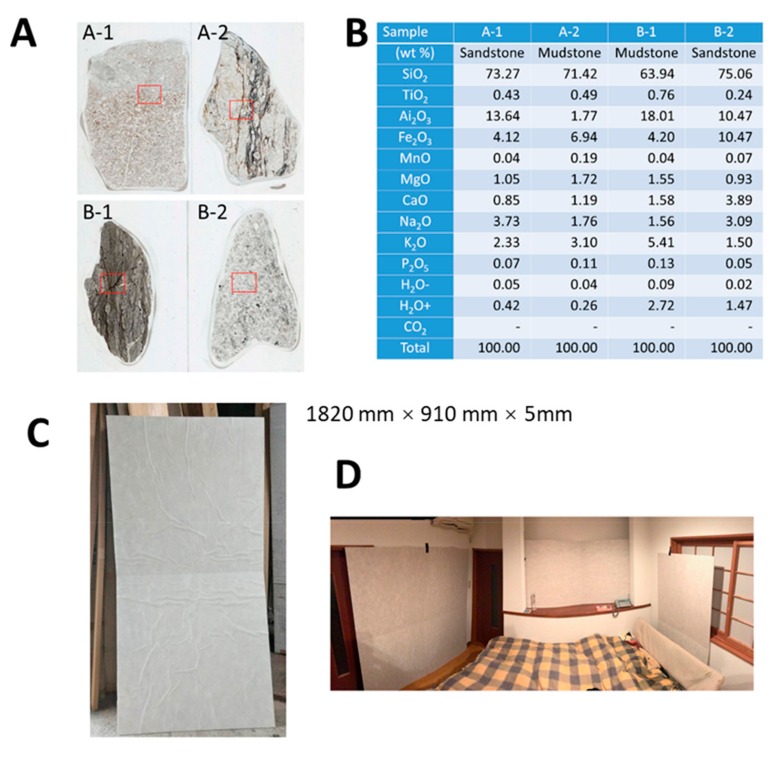
The specific natural ore and placement of cloth containing specific natural ore powder (CCSNOP). (**A**) Appearance of the natural ore obtained near Aso Mountain located in Kumamoto Prefecture, Kyushu Island, Japan. Labels A and B show the mirror-polished flake appearance (normal light) of this natural ore. The red square is where the photomicrograph was taken for analysis. The width of the slide glass is 28 mm. A-1 is sandstone, A-2 is mudstone, B-1 is mudstone, and B-1 is sandstone. (**B**) Results of the chemical analysis using X-ray fluorescence of four parts of the natural ore (A-1, A-2, B-1, and B-2). None of these showed any particularly characteristic findings. (**C**) Appearance and size of CCSNOP panels used in this experiment. Non-woven cloth (NWC) panels had a similar appearance. Each panel was folded and four panels were delivered to each subject. Subjects were responsible for placement of the panels within their bedroom. (**D**) Picture of a bedroom containing panels (subject ID 60, posted with permission of the subject).

**Figure 2 jcm-08-02164-f002:**
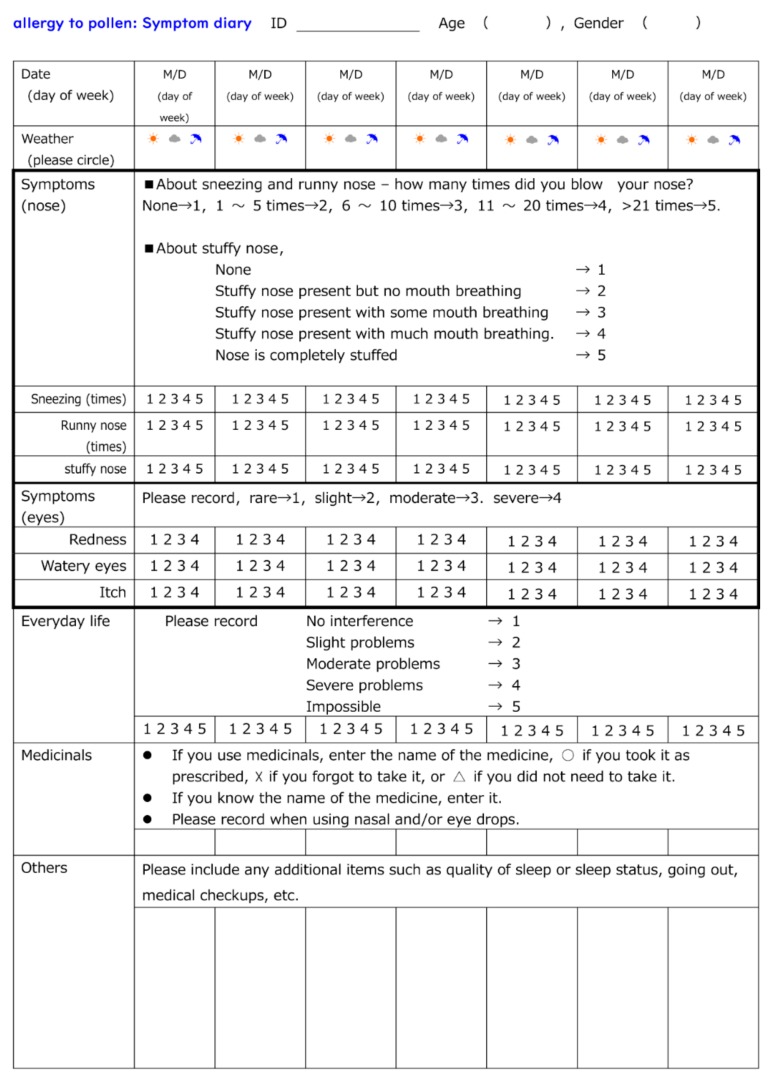
Pollen allergy diary. Each subject was asked to keep a diary. In addition to the date, day of the week, and an assessment of the weather, subjects also recorded any symptoms related to the nose or eye on a scale of 1–5. Subjects were also asked to describe the difficulty or impact of their symptoms on their daily life. Additionally, the use of medication (oral drugs, nasal drops, and eye drops) was described. Moreover, subjects could list any physical or mental changes they may have noticed. Diary entries were recorded after commencement of pollen dispersal, from one week prior to panel placement, for two weeks during panel exposure, after panel removal, and the one week where pollen dispersal continued (for a total of four weeks).

**Figure 3 jcm-08-02164-f003:**
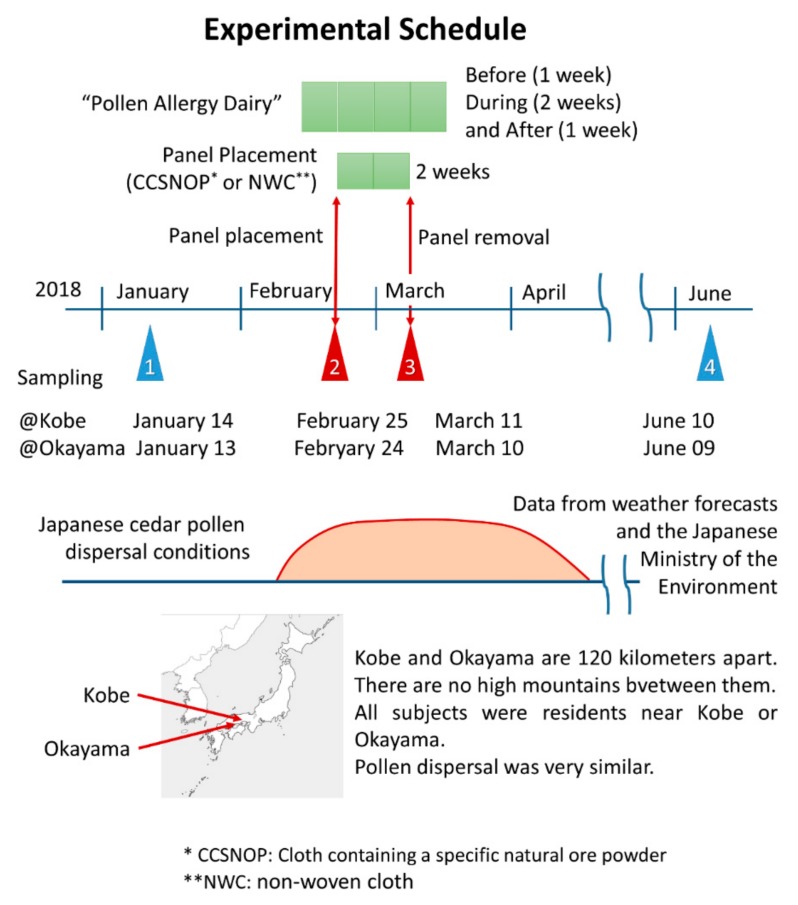
Experimental schedule. The study was performed from January to June 2018. According to data from Japanese weather forecasts and the Japanese Ministry of the Environment, Japanese cedar pollen dispersal began in early February and ceased at end of April 2018. The degree of pollen dispersal in the Kobe and Okayama areas (where all subjects of this study were living) was similar during this period since both cities are located in the western region of Japan, as shown in the map, and are located ca. 120 km apart. There are no high mountains between these areas. The sampling was performed four times. The data (mood assessment by POMS2, stress marker by salivary amylase, and blood collection analyses) were collected on January 13 or 14, February 24 or 25, March 10 or 11, and June 9 or 10 at the Okayama and Kobe areas, respectively. The first and last samplings were obtained and represent data reflecting the absence of pollen dispersal. CCSNOP or NWC panels were placed just after the second sampling up until and immediately after the third sampling. Subjects were asked to keep a “pollen allergy diary” as described in the caption of Figure 2.

**Figure 4 jcm-08-02164-f004:**
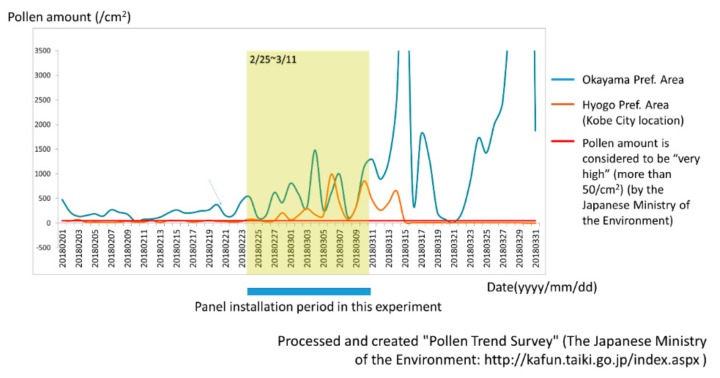
Changes in the amount of pollen during the panel installation period in Okayama City (Okayama Prefecture, Blue Line) and Kobe City (Hyogo Prefecture, Orange Line) where the subjects of this study were located, as obtained from the pollen trend survey results of the Japanese Ministry of the Environment [13]. The red line indicates a pollen level of 50/cm^2^, and represents a “very high” level according to the Ministry of the Environment of Japan.

**Figure 5 jcm-08-02164-f005:**
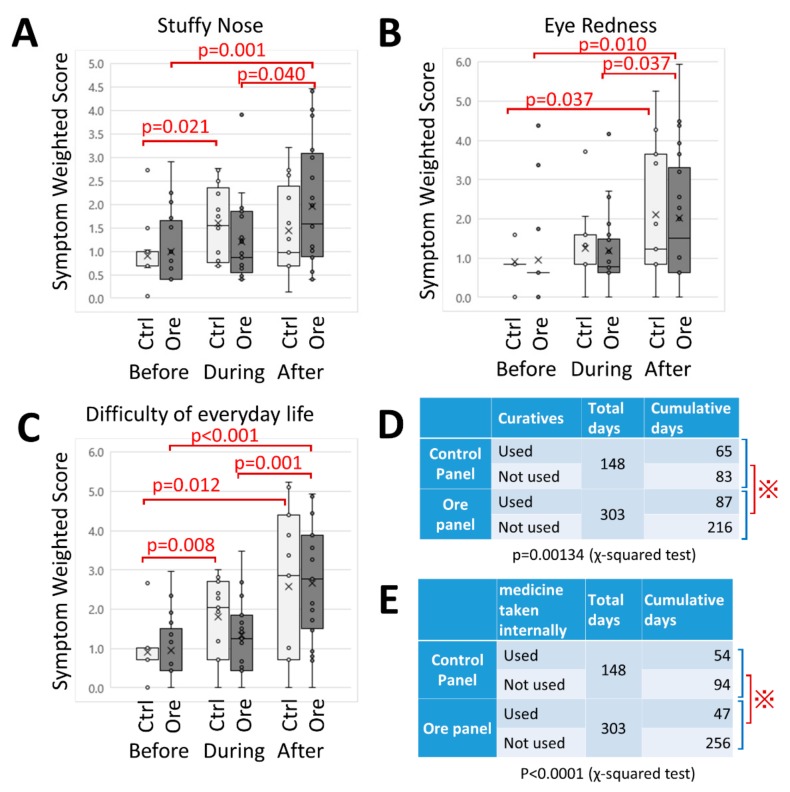
Changes in symptoms. (**A**) stuffy nose, (**B**) redness of eyes, and (**C**) difficulty of everyday life. Data were obtained from the “pollen allergy diary” of individual subjects. Values of severity ranging from 1 to 5 were squared (as “weighted”) to clarify the differences. Panels A to C show box-plot graphs. White and gray boxes represent data from NWC (control; ctrl) and CCSNOP (ore) subjects, respectively. “Before”, “During”, and “After” indicate one week before panel placement, two weeks of panel exposure, and one week after panel removal, respectively. The red lines indicate significant difference (*p* < 0.05). (**D**) and (**E**) show the results of a *χ*-squared test. (**D**) Use of allergy-related medicines. (**E**) Use of medicine taken internally. “°” and “×” indicsate individual data and average, respectively. “⁜” showed statistical significance.

**Figure 6 jcm-08-02164-f006:**
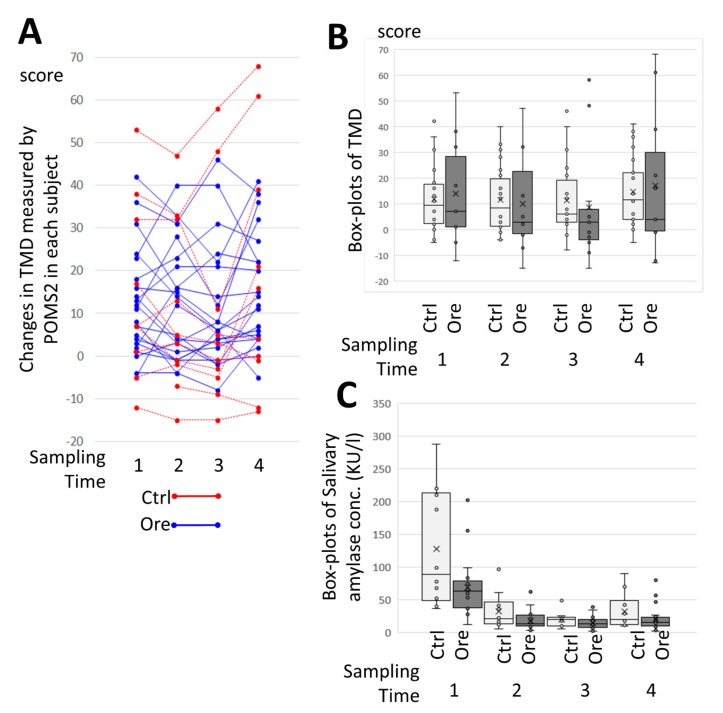
Results of total mood disturbance (TMD) and stress marker. (**A**) Actual changes in TMD assayed by Survey of Mood (POMS)2 in all subjects during sampling times 1–4. The red and blue lines indicate NWC (ctrl) and CCSNOP (ore) subjects, respectively. A varieties of changes were observed with no regular pattern being observed. (**B**) Comparison of TMD in the NWC (ctrl) and CCSNOP (ore) groups at sampling times 1 to 4. There was no significant difference between the two groups. (**C**) Box-plots show changes in salivary amylase levels (sAmy) as a stress marker in the CCSNOP (ore) and NWC (ctrl) groups during sampling times 1–4. There was no significant difference between the two groups.

**Figure 7 jcm-08-02164-f007:**
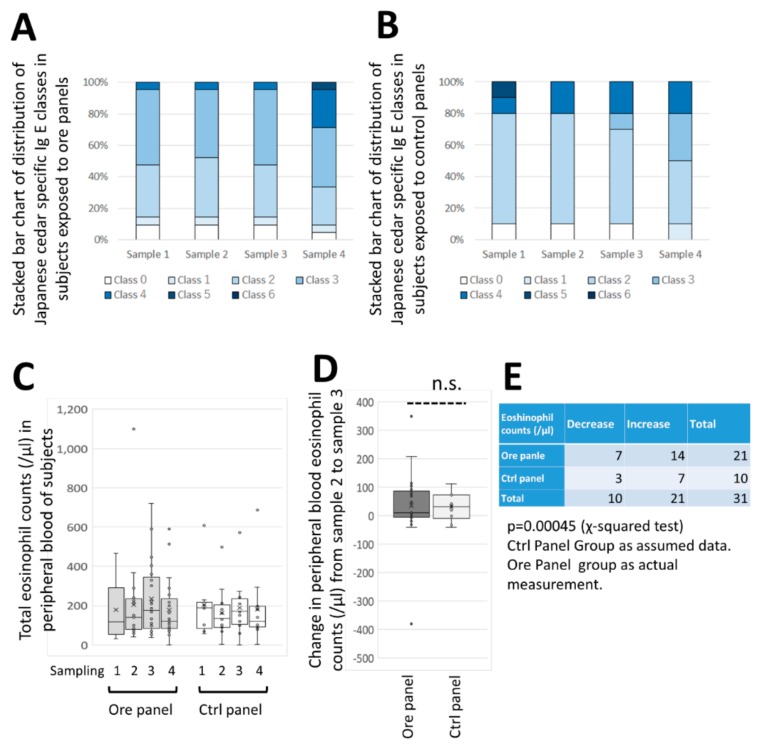
Changes in cedar pollen-specific Ig E and eosinophils. Stacked bar charts showing the distribution of Japanese cedar specific Ig E classes 0–6 among (**A**) CCSNOP (ore) and (**B**) NWC (control) subjects. (**C**) Box-plots of total peripheral eosinophil counts in CCNSOP (ore) and NWC (ctrl) subjects during sampling times 1 to 4. There were no significant differences between the two groups. (**D**) Box-plots of peripheral blood eosinophil counts from sampling times 2–3 in the CCSNOP (ore) and NWC (ctrl) groups. There were no differences found. (**E**) χ-squared test of subjects who showed an accelerated decrease in peripheral blood eosinophil counts from sampling times 2–3. There was a significant difference and the CCSNOP (ore) group showed an accelerated decrease in eosinophils. “n.s.” indicate “not significant”.

**Figure 8 jcm-08-02164-f008:**
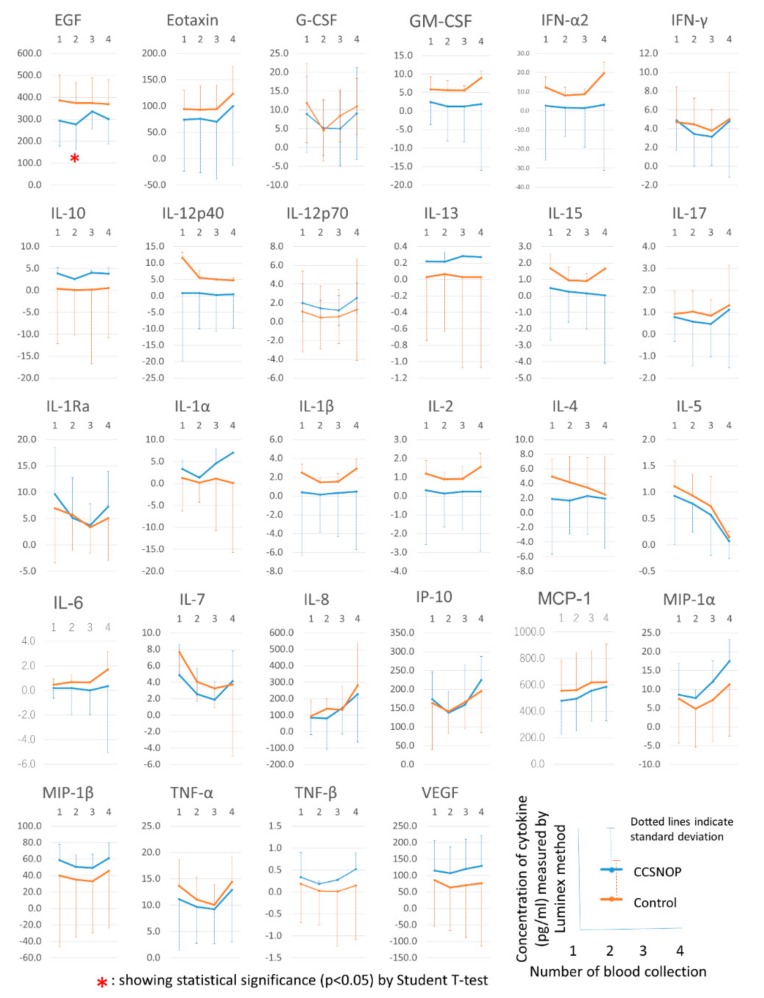
Twenty-eight kinds of cytokines (interleukin 3 (IL-3) was excluded because it could only be measured with one sample) were measured in four blood collections, and the average value and standard deviation are shown for the CCSNOP and control groups. Moreover, a comparison between each group by Student’s t-test revealed that the CCSNOP group showed a higher value than the control group in the first and second blood collection of epidermal growth factor (EGF). G-CSF: granulocyte-colony stimulating factor; granulocyte-macrophage-colony stimulating factor; GM-CSF. IFN: interferon; MCP: monocyte chemotactic protein-1; IP-10: interferon γ-induced protein 10; MIP: macrophage inflammatory protein; TNF: tumor necrosis factor; VEGF: vascular endothelial growth factor.

**Figure 9 jcm-08-02164-f009:**
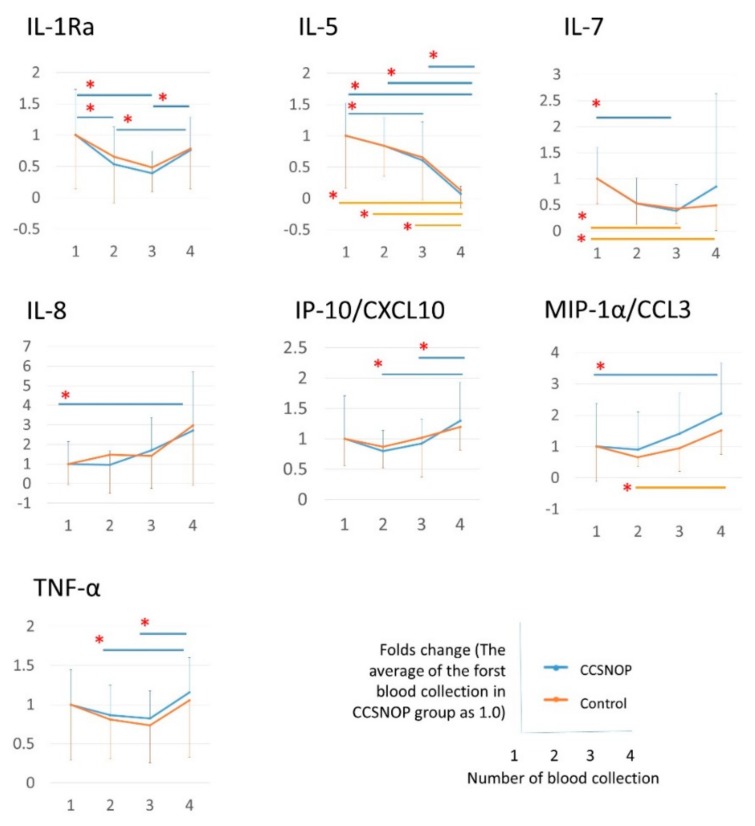
The changes over time of 28 cytokines are shown as a result of examining the relative changes with the average value of the first blood collection in the CCSNOP group as 1.0. Among the 28 cytokines whose real numbers are shown in Figure 8, seven cytokines that showed a significant difference in the time course of the CCSNOP group or the control group are shown. “*“ indicates statistical significance (p<0.05).

**Figure 10 jcm-08-02164-f010:**
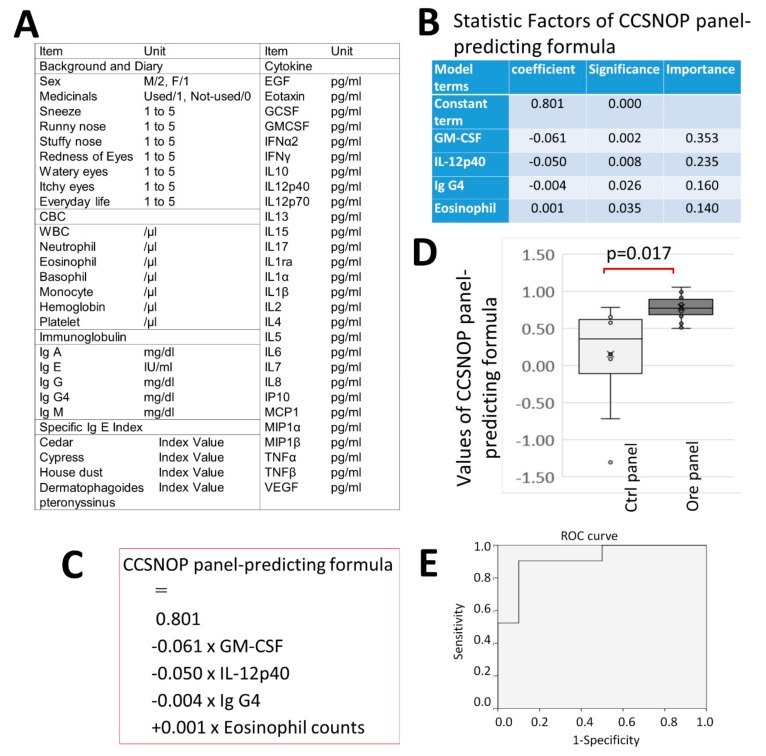
Formula predicting subjects exposed to CCSNOP (ore). (**A**) List of parameters employed for multiple regression analysis. (**B**) Factors, coefficient, significance, and importance of statistical factors extracted in CCSNOP panel-predicting formula. (**C**) The formula for CCSNOP panel prediction. (**D**) Using the CCSNOP panel-predicting formula, all data pertaining to GM-CSF, IL-12p40, IgG4, and eosinophil counts at sampling time 3 were substituted. Thereafter, values derived from the formula for CCSNOP (ore) and NWC (ctrl) subjects were plotted in the form of box-plots. A significant difference (*p* = 0.017) was observed. (**E**) Receiver Operatorating Characteristic (ROC) curve of the CCSNOP panel-predicting formula, with (sensitivity) and (1-specificity) values of 0.905 and 0.100, respectively.

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
