# Peer review of "Effects of a Cloth Panel Containing a Specific Ore Powder on Patients with Japanese Cedar Pollen Allergy During the Pollen Dispersal Season"

_jcm, 2019, doi:10.3390/jcm8122164_

Round 1

Reviewer 1 Report

The authors have adequately addressed the reviewer's comments. I do not have additional comments. The manuscript is now suitable for publication.

Author Response

Would you please see attached file?

Reviewer 2 Report

In this manuscript Lee and colleagues demonstrated a correlation between the use of the specific ore panels and the alleviation of the pollen allergic symptoms.

The manuscript is well written and the methods are sufficiently described. The results section could be improved by emphasizing the area where there is a significant difference between the groups (like the stuffy nose).

Also, in the Figure 8 depicting cytokine measurements the data are expressed in negative values. Is there a way to present the data in a different way, e.g. fold change?

The authors could discuss the difference observed for the stuffy nose group in the context of before and after exposure. Also why in authors' opinion they did not observe the difference in other symptoms?

Finally, the authors do not propose a mechanism of how the ore panels could mitigate the allergy symptoms.  This should be addressed in the discussion section.

Author Response

Would you please see attached file?

This manuscript is a resubmission of an earlier submission. The following is a list of the peer review reports and author responses from that submission.

Round 1

Reviewer 1 Report

Lee et al has submitted research article entitled “Effects of cloth panel containing specific ore powder…… dispersal season” for publication in Journal of Clinical Medicine.

The authors have performed experiments on 31 subjects using either cloth containing a special natural ore powder (CCSNOP) or non-woven cloth (as control) to understand the effect of cloth containing natural ore powder on pollen allergy. The manuscript lacks a substantial amount of data that really can confirm the effect of CCSNOP in reducing allergy reaction and immune modulation in pollen exposed subjects.

Major concerns:

The study design has a lot of flaws. It is not known how long the subjects are exposed to the outside environment of the house where pollens are prevalent. Also not know how much pollens are present inside the house.

The authors have mentioned that pollen dispersal is similar in both Kobe and Okayama. But no data has been presented in the manuscript.

The authors have emphasized more on the subjective data like mood survey and symptoms diary. However, I believe that major emphasize should be given to look over detailed proinflammatory, inflammatory and activation markers in different T and B cell subsets that can be generated from blood samples collected during different time point of this study.

I am against the formula for measuring various cytokines and other parameters. That formula needs to be validated in some animal model, where the animals should be exposed with pollens and will be compared with controls (no pollen exposed), which has not been presented here.

Moreover, only cytokine data from sampling time 3 have been used for comparison between CCSNOP and NWC groups. The dynamics of each cytokine/chemokine data are missing and need to be shown in this manuscript. The activation and proliferation markers in the peripheral blood are also missing.

No discussion has been shown for several important cytokines like MCP1, IP10, CXCL10, CCL4 and CCL3 that are regulator for allergic reactions.

Lastly the paper is poorly written.

Minor comments

So many spelling errors throughout the manuscript.

There is no figure legend for figure 7. Both Fig.6 and 7 have same figure legend.

Reviewer 2 Report

In that study, authors attempted to evaluate the effect exposure of cloth containing a special natural ore powder (CCSNOP) on individuals with pollen allergy. The study subjects were exposed to CCSNOP or control (non-woven cloth; NWC) by using cloth panels placed in the bedrooms of pollen allergy patients for two weeks during the pollen dispersal season.  Here they monitored several biological and psychological parameters related to pollen allergy symptoms. Overall, they found that exposure to CCSNOP alleviated the symptoms of pollen allergy, however, several parameters of allergic responses were similar between the CCSNOP exposed and the control group. Finally, the authors measured serum cytokines form the study subjects. Then they attempted to derive a formula based on several measured parameters to predict the improvement of allergic response.

Overall, the manuscript is well written and the text flows well. The methods are clearly mentioned and well elaborated, and the conclusions are based on appropriate results.

Minor comment:

In Figure 4, it would be appropriate if the authors indicate appropriate P values for statistical comparison between different groups. P values can be added above the horizontal red brackets.

Reviewer 3 Report

In my opinion the paper is an interesting study on pollen allergy, being an ongoing problem in may populations. I think that the Intro is correct, informative and prompt. The biggest value of the paper is Methods - a great panel of methods was used and showed in results. It is interpreted correctly. I recommend the paper to be printed in its current form.